PhD7Faster 2.0: predicting clones propagating faster from the Ph.D.-7 phage display library by coupling PseAAC and tripeptide composition

He Bifang 1 2
http://orcid.org/0000-0001-6939-7665 Chen Heng 1
Huang Jian 2 hj@uestc.edu.cn
1 School of Medicine, Guizhou University , Guiyang, Guizhou , China
2 Center for Informational Biology, University of Electronic Science and Technology of China , Chengdu, Sichuan , China
Tatarinova Tatiana
Electronic publication date: 2019 Jun 17
Publication date: 2019
Volume: 7
Electronic Location ID: e7131
Received 2018 Dec 22; Accepted 2019 May 15
Copyright: © 2019 He et al.
Copyright year: 2019
Copyright holder: He et al.
License: This is an open access article distributed under the terms of the Creative Commons Attribution License, which permits unrestricted use, distribution, reproduction and adaptation in any medium and for any purpose provided that it is properly attributed. For attribution, the original author(s), title, publication source (PeerJ) and either DOI or URL of the article must be cited.
License URL: https://creativecommons.org/licenses/by/4.0/

Keywords: Propagation-related TUPs, Ph.D.-7 phage display library, Predictor, Feature selection, Support vector machine

Funding: National Natural Science Foundation of China 61571095 Fundamental Research Funds for the Central Universities of China ZYGX2005Z006 Sichuan Science and Technology Program 2018HH0154 2018 Talent Research Program of Guizhou University 702569183301 and 702570183301 Science and Technology Plan Project of Guizhou Province of China (2018)5781 China Postdoctoral Science Foundation Grant 2019M653369 This work was supported by the National Natural Science Foundation of China (grant number 61571095), the Fundamental Research Funds for the Central Universities of China (ZYGX2005Z006), the Sichuan Science and Technology Program (2018HH0154), the 2018 Talent Research Program of Guizhou University (702569183301 and 702570183301), the Science and Technology Plan Project of Guizhou Province of China ((2018)5781), and the China Postdoctoral Science Foundation Grant (Grant No. 2019M653369). The funders had no role in study design, data collection and analysis, decision to publish, or preparation of the manuscript.

==============================
Selection from phage display libraries empowers isolation of high-affinity ligands for various targets. However, this method also identifies propagation-related target-unrelated peptides (PrTUPs). These false positive hits appear because of their amplification advantages. In this report, we present PhD7Faster 2.0 for predicting fast-propagating clones from the Ph.D.-7 phage display library, which was developed based on the support vector machine. Feature selection was performed against PseAAC and tripeptide composition using the incremental feature selection method. Ten-fold cross-validation results show that PhD7Faster 2.0 succeeds a decent performance with the accuracy of 81.84%, the Matthews correlation coefficient of 0.64 and the area under the ROC curve of 0.90. The permutation test with 1,000 shuffles resulted in p < 0.001. We implemented PhD7Faster 2.0 into a publicly accessible web tool (http://i.uestc.edu.cn/sarotup3/cgi-bin/PhD7Faster.pl) and constructed standalone graphical user interface and command-line versions for different systems. The standalone PhD7Faster 2.0 is able to detect PrTUPs within small datasets as well as large-scale datasets. This makes PhD7Faster 2.0 an enhanced and powerful tool for scanning and reporting faster-growing clones from the Ph.D.-7 phage display library.

Introduction

Phage display is a high throughput and powerful screening methodology for identifying ligands for myriad target types, ranging from molecules (microRNA, protein, polysaccharide) (He et al., 2013; Zhang et al., 2017) to inorganic (gold) (Causa et al., 2013), organic (epoxy) (Swaminathan & Cui, 2013), and biological (tissue, organ) materials (Hung et al., 2018). Large libraries of phage-displayed peptides or proteins consist of millions to billions of variant members, which can be iteratively selected and amplified in a process referred to as biopanning (Pande, Szewczyk & Grover, 2010). Recently, next generation sequencing technologies have been coupled with phage display, which have substantially contributed to the analysis of output from combinatorial libraries and allowed for even faster and more robust discovery of novel ligands (Christiansen et al., 2015; Matochko et al., 2014; Ngubane et al., 2013; Rentero Rebollo et al., 2014; ‘t Hoen et al., 2012). The ever-increasing utility and versatility makes phage display a powerful tool in multiple research areas, such as materials science, biotechnology, pharmacology, cell biology, and diagnostics (Martins, Reis & Azevedo, 2016).

However, the phage display methodology is notorious for the enrichment of target-unrelated peptides (TUPs) (Menendez & Scott, 2005). Therefore, biopanning results are a mixture of true target binders and TUPs (Vodnik et al., 2011). These false positive TUPs have no actual affinity toward the target of interest and can fall into two categories: selection- and propagation-related TUPs (SrTUPs and PrTUPs) (Thomas, Golomb & Smith, 2010). The SrTUPs can bind to other components (plates, beads) of the screening system other than the desired target and thus creep into the output of phage display. The PrTUPs sneak into the biopanning results due to their propagation advantages, which allow them to outcompete clones with lower growth rates (Brammer et al., 2008; Matochko et al., 2014; Nguyen et al., 2014; Thomas, Golomb & Smith, 2010; Zade et al., 2017; Zygiel et al., 2017). Apparently, these TUPs may misdirect ligand discovery through biopanning and should be distinguished from actual target-binding peptides (Bakhshinejad et al., 2016). Therefore, the diagnosis of TUPs is as crucial as the identification of target binders.

Although several experimental strategies have been proposed to decrease TUP isolation during biopanning and differentiate between TUPs and true binders post-biopanning (Nguyen et al., 2014; Thomas, Golomb & Smith, 2010; Vodnik et al., 2011), TUP analysis has benefitted considerably from computational approaches. Databases (BDB (He et al., 2016a, 2018; Huang et al., 2012; Ru et al., 2010), PepBank (Shtatland et al., 2007)) and bioinformatics tools (He et al., 2016b; Huang et al., 2010; Li et al., 2017; Mandava et al., 2004; Ru et al., 2014) have been widely employed to report both SrTUPs and PrTUPs. Searching against databases for biopanning data can uncover whether query peptides have been isolated by many different targets. If so, query sequences are potential SrTUPs and PrTUPs due to lack of target specificity. For example, the peptide HAIYPRH (a typical PrTUP) has been identified by 23 completely different targets according to results of searching the BDB database. The phage displaying the peptide was later verified to have a propagation advantage owing to mutations in the regulatory region of the phage genome (Brammer et al., 2008). HWGMWSY (a SrTUP) has been isolated by 10 completely different targets according to records in the BDB database. The peptide was proved to be a plastic binder (Vodnik, Strukelj & Lunder, 2012), which resulted in this peptide repeatedly appearing in multiple reported screening experiments. SABinder (He et al., 2016b) and PSBinder (Li et al., 2017) have been designed for predicting streptavidin- and polystyrene surface-binding peptides, respectively, as they are commonly known SrTUPs. The INFO tool in the RELIC suite enables PrTUPs detection based on information content (Mandava et al., 2004), whereas PhD7Faster (PhD7Faster 1.0) based on support vector machine (SVM) allows the prediction of clones with amplification advantages from the popular commercial Ph.D.-7 phage display library (Ru et al., 2014). However, PhD7Faster 1.0 can be improved in the following three aspects. Firstly, the positive training dataset of PhD7Faster 1.0 was selected based on the copy number of a peptide (15 or higher) after one round of amplification without consideration of the corresponding copy number in the naïve Ph.D.-7 library. Secondly, only dipeptide composition was employed to develop the classifier. Currently, many reports have demonstrated that predictors developed by combining pseudo amino acid composition (PseAAC) (Chou, 2001, 2005) and tripeptide composition can achieve decent predictive performances (Liao et al., 2011; Zhu et al., 2015). Thirdly, PhD7Faster 1.0 is unable to process large datasets (e.g., next-generation sequencing data).

In this study, we develop a new predictor for identifying clones propagating faster from the Ph.D.-7 phage display library. The SVM algorithm was employed to model the predictor with the optimal feature subset after feature selection. The constructed SVM-based classifier obtained an accuracy of 81.84% in the ten-fold cross-validation. The predictor was further implemented into a web tool, called PhD7Faster 2.0, which is freely available at http://i.uestc.edu.cn/sarotup3/cgi-bin/PhD7Faster.pl. We also developed the standalone version of PhD7Faster 2.0 that enables the analysis of PrTUPs within large-scale datasets.

Data and Methods

Benchmark datasets

The dataset used to develop the predictor was acquired from (Matochko et al., 2014). Derda et al. employed high-throughput sequencing technology to characterize both the naïve Ph.D.-7 phage display library and the same library after one round of amplification. By comparing the abundance of each peptide before and after amplification using Bioconductor package edgeR, 770 unique peptides were identified with significantly higher growth rate (parasitic sequences) (Matochko et al., 2014), which were collected into the positive training dataset. The negative dataset was composed of those peptides with the copy number of one in the amplified Ph.D.-7 phage display library. The datasets were then processed as follows: (i) peptide sequences containing ambiguous residues (such as “X”, “B,” and “Z”) were excluded; (ii) sequences within 2 Hamming distance (h = 2, the Hamming distance between two strings of equal length is the minimum number of substitutions required to change one string into the other.) were removed. Finally, 749 peptides were retained in the positive dataset. To match the size of the positive dataset, we randomly selected 749 peptides from the negative dataset. No overlapping was found between the negative and positive datasets. Finally, the benchmark dataset was composed of 749 fast-growing peptides and 749 regular-growing peptides (See positive.fasta and negative.fasta in Supplementary Data).

PseAAC and tripeptide composition

Extracting a set of informative features is a standard and important procedure for developing predictors. Chou initially formulated the PseAAC (Chou, 2001, 2005), which consists of more than 20 discrete numbers, where the top 20 represent the classical amino acid composition (AAC) of a protein sequence whereas the additional parameters incorporate some sequence-order information. PseAAC and tripeptide composition have been widely used in protein prediction related research (Chou, 2011; Lin et al., 2013). Here, they were employed to encode each peptide in the benchmark dataset.

Given a peptide P with L amino acid residues:(1) P=(R1R2R3R4R5R6R7…RL)

where Ri (i = 1, 2, 3 . . . L) is the residue at the ith sequence position. Accordingly, any sequence like the peptide P of Eq. (1) can be presented using a set of feature vectors with 8,000 + nλ dimensions.

(2) P=(P1,P2,⋯,P8,000,P8,000+1,⋯,P8,000+nλ)

where the first 8,000 numbers P1, P2, . . ., P8,000 reflect the effect of the conventional tripeptide composition; the remaining nλ elements P8,000+1, P8,000+2, . . ., P8,000+nλ reflect the amphipathic sequence-order pattern. These features are calculated through the following equations:(3) Pu={fu∑i=18,000fi+w∑j=1nλτj(1≤u≤8,000)wτu∑i=18,000fi+w∑j=1nλτj(8,000+1≤u≤8,000+nλ)

where fi (i = 1, 2, 3, . . ., 8,000) are the normalized occurrence frequencies of the 8,000 tripeptides in peptide P; w is the weight factor for the sequence-order effect; (τj (j = 1, 2, . . ., nλ) is the j-tier sequence-correlated factor as formulated by:(4) {τ1= 1L−1∑i=1L−1Hi,i+11τ2= 1L−1∑i=1L−1Hi,i+12…τn=1L−1∑i=1L−1Hi,i+1nτn+1=1L−2∑i=1L−2Hi,i+21τn+2=1L−2∑i=1L−2Hi,i+22…τ2n=1L−2∑i=1L−2Hi,i+2n…τnλ−1=1L−λ∑i=1L−λHi,i+λn−1τnλ=1L−λ∑i=1L−λHi,i+λn

where Hi,jn is the physicochemical property correlation function and can be computed according to the following equation:(5) Hi,jn=hn(Ri).hn(Rj)

where hn(Ri) and hn(Rj) are the values of the nth type of physicochemical property of Ri and Rj in Eq. (1), respectively. It is noteworthy that before substituting the values of all physicochemical properties into Eq. (5), they were undergone a standard conversion as described below:(6) hk(Ri)=h0k(Ri)−∑α=120h0k(Rα)/20∑u=120(h0k(Ri)−∑α=120h0k(Rα)/20)2

where Ri (i = 1, 2, . . ., 20) denotes the 20-standard amino acid in the alphabetical order of their single-letter codes. h0k(Ri) is the initial value of the kth type of physicochemical property for amino acid residue Ri. Nine kinds of physicochemical properties, namely hydrophobicity, hydrophilicity, mass, pK1, pK2, pI, rigidity, flexibility, and irreplaceability, were considered in this report.

Feature selection

Generally, not all features make an equal contribution to the prediction system. A part of features make significant contributions, while some others make less important contributions (Zhao et al., 2016). Feature selection, thus, is a critical step to reduce feature dimensionality and build a highly effective prediction model (Su et al., 2018; Tang, Chen & Lin, 2016). In this work, the fselect.py program in the LIBSVM 3.23 package was applied to evaluate each feature’s significance to the classification system (Chang & Lin, 2011). As a consequence, each feature corresponds to an F-score. The greater F-score implies the larger importance of the corresponding feature to the prediction model. We rearranged all features by F-scores in descending order. The incremental feature selection strategy was then utilized to determine the optimal feature subset (He et al., 2016b; Li et al., 2017), which can produce the maximal accuracy. Feature selection was conducted as follows: (i) investigating the accuracy of the first feature subset which included the feature with the largest F-score; (ii) examining the accuracy of the second feature subset that was generated by appending the feature with the second largest F-score; (iii) iterating the second step from the larger F-score to the smaller F-score until all candidate features were added. The best feature subset with the highest accuracy can be finally obtained.

Support vector machine

The SVM is a powerful supervised learning method, which has been widely applied in classification (He et al., 2016b; Kang et al., 2018; Li et al., 2017; Ru et al., 2014) and regression analysis. In this study, we utilized the LIBSVM 3.23 program (Chang & Lin, 2011) that could be freely available for download from http://www.csie.ntu.edu.tw/∼cjlin/libsvm/. We chose the radial basis function kernel as the kernel function. The optimal kernel width parameter γ and penalty constant C were selected by using the parameter selection tool in the LIBSVM 3.23 (Chang & Lin, 2011).

Performance evaluation

The 10-fold cross-validation was adopted to evaluate the predictive model in this study. Four commonly-used parameters, including sensitivity (Sn), specificity (Sp), accuracy (Acc), and Matthews correlation coefficient (MCC), were employed to investigate the performance of the constructed model. These measures were expressed as follows:(7) Sn=TPTP+FN

(8) Sp=TNFP+TN

(9) Acc=TP+TNTP+FN+FP+TN

(10) MCC=TP×TN−FP×FN(TP+FP)(TP+FN)(TN+FP)(TN+FN)

where TP and TN denote the number of true positives and negatives, respectively. FP and FN are the number of false positives and negatives, respectively. The area under the receiver operating characteristic (ROC) curve (AUC) was also calculated as a performance measure. The AUC ranges from zero to one. The AUC of one represents a perfect prediction, 0.5 a random guess.

To estimate the statistical significance of the predictive accuracy, a permutation test with 1,000 shuffles was performed by exchanging the labels of the benchmark dataset. The 10-fold cross-validation was then conducted against the label-rearranged dataset. Thus, each permutation trial corresponds to an accuracy value. The p-value was calculated by the number of permutations that the Acc produced by the permuted dataset was higher than Acc based on the un-permuted dataset divided by the overall shuffle times. p-values of <0.05 were referred to as statistically significant.

Standalone version implementation

The standalone version of PhD7Faster 2.0 was developed with open source Qt 5.7 under the GPL & LGPLv3 licenses, which uses standard C++ for developing multiple-platform applications. Both graphical user interface (GUI) and command-line versions of PhD7Faster 2.0 were implemented. We provided different versions for Windows and Linux systems with little or no modification. All versions and source code are freely available at http://i.uestc.edu.cn/sarotup3/download.html.

Results

Parameter optimization

Two important parameters: λ and w in Eq. (3) were necessary to be optimized before building the model. To obtain the best parameters, multiple experiments were performed according to the following standard:(11) {1≤λ≤6 with step Δ=10.05≤w≤0.70 with step Δ=0.05

Thus, a total of 6 × 14 = 84 individual combinations were obtained. Then, we used the 10-fold cross-validation to investigate the accuracy of the model, which was built with SVM and the feature set without feature selection. λ = 3 and w = 0.15 produced the highest accuracy, which was considered as the best parameter combination.

Performance of PhD7Faster 2.0

The optimal feature subset with 644 features was determined through feature selection against 8,027 features including 8,000 tripeptide features and 27 PseAAC features. The SVM-based model was then trained with the optimal feature set. The results from the 10-fold cross-validation showed that the Acc of the predictive model was 81.84% with MCC of 0.64, Sn of 84.51%, and Sp of 79.17% when the threshold to distinguish between predicted positives and negatives (tp) was set to be 0.5. The ROC curve for model tuning is shown in Fig. 1, where the AUC is approximately 0.90. The permutation test resulted in a p-value of < 0.001. The above results indicated that PhD7Faster 2.0 achieved a promising performance.

Figure 1 The ROC curve from the 10-fold cross-validation when tp is 0.5.

The area under the ROC curve is about 0.90, which represents a decent prediction.

Web and standalone versions of PhD7Faster 2.0

For the convenience of users, the SVM-based predictive model was implemented into a user-friendly web server, called PhD7Faster 2.0, which is freely available at http://i.uestc.edu.cn/sarotup3/cgi-bin/PhD7Faster.pl. The standalone GUI and command-line versions of PhD7Faster 2.0 for Windows and Linux systems were also provided. The interface, as well as the utilization of the GUI version, is remarkably similar to those of the web version (Fig. 2). A dataset with 20,000 peptides from the Ph.D.-7 phage display library was constructed (see testdataset.fasta in Supplementary Data). The standalone PhD7Faster 2.0 can complete analysis of the dataset within 60 s on a regular computer with Intel Core i3 Processor and 4GB RAM, which suggests that PhD7Faster 2.0 is highly efficient in processing massive datasets. PhD7Faster 2.0 was integrated into the SAROTUP 3.0 suite, which contains a series of computational tools to identify TUPs.

Figure 2 GUI version, web interface and output interface of PhD7Faster 2.0.

The interface style of the GUI PhD7Faster 2.0 is consistent with that of the web server, which makes the tool user-friendly.

Discussion

Comparison between PhDFaster 2.0 and 1.0

Parasitic sequences were identified significantly enriched in the amplified Ph.D.-7 phage display library by differential enrichment analysis of naïve and amplified Ph.D.-7 phage display libraries (Matochko et al., 2014). These parasitic peptides were grouped into the positive dataset of PhD7Faster 2.0. However, the positive dataset of PhD7Faster 1.0 was constructed based on threshold in copy numbers after one round of amplification in one replicate of sequencing data, irrespective of copy numbers in the naïve Ph.D.-7 library. Peptides with high abundances in both the naïve and amplified Ph.D.-7 libraries may also be selected as fast-growing sequences. Therefore, the positive training dataset of PhD7Faster 2.0 is more reliable than that of PhD7Faster 1.0.

PhD7Faster 2.0 was developed based on the combination of PseAAC and tripeptide composition, whereas only dipeptide composition was employed to build PhD7Faster 1.0. We also tried to use dipeptide composition to encode each peptide in the training dataset of PhD7Faster 2.0, but only 64% accuracy was obtained in the 10-fold cross-validation after feature selection. PseAAC coupled with tripeptide composition has been used in multiple protein prediction fields, such as predicting the subcellular localization of mycobacterial proteins (Zhu et al., 2015) and predicting apoptosis protein subcellular location (Liao et al., 2011). They contain more sequence-order information than dipeptide composition and hence can better reflect the feature of a peptide sequence. Thus, PhD7Faster 2.0 has 5% sensitivity, 2% accuracy and 0.04 MCC higher than PhD7Faster 1.0 (Table 1).

Table 1 Comparison of performances of PhD7Faster 1.0 and 2.0.

Method	Sn (%)	Sp (%)	Acc (%)	MCC	
PhD7Faster 1.0	77.48	81.86	79.67	0.60	
PhD7Faster 2.0	84.51	79.17	81.84	0.64	
Note:

The measure of PhD7Faster 2.0 higher than that of PhD7Faster 1.0 is highlighted in bold.

The standalone PhD7Faster 2.0 is empowered to identify PrTUPs within output of conventional phage display as well as large next-generation sequencing data, whereas PhD7Faster 1.0 can only work with small-scale data sets (several hundreds of peptides). This important improvement makes PhD7Faster 2.0 as an enhanced and powerful tool for scanning and reporting PrTUPs from the Ph.D.-7 phage display library. The emergence of PhD7Faster 2.0 highlights the significance of high throughput sequencing of different types of phage display libraries and developing bioinformatics tools for identifying PrTUPs from these libraries.

PhD7Faster 2.0 cannot predict the censorship in the Ph.D. libraries

It is possible that some peptides are likely to be censored from being displayed on the phage in the first place. The censorship of positively charged amino acids has been reported since these residues suppress proper insertion of pIII into the inner membrane of Escherichia coli, thus decreasing the efficiency of the assembly and extrusion of phage clones (Peters et al., 1994). Rodi, Soares & Makowski (2002) also observed that peptides of α-helix or β-sheet conformations were censored in Ph.D.-12 and Ph.D.-C7C libraries. Steiner et al. (2006) have shown that maturely folded proteins are displayed poorly via the Sec translocation pathway. However, this censorship is a completely different phenomenon from that of phage growing faster. Therefore, PhD7Faster 2.0 is not able to predict this censorship.

PhD7Faster 2.0 predict PrTUPs in the Ph.D.-7 library

The PrTUPs have significantly higher proliferation rates than normal-growing phage and are favored during the amplification steps. The proliferation advantage of some PrTUPs have been verified to be intrinsic to mutations in the 5′-untranslated region (UTR) of gene II in M13 phage (Brammer et al., 2008; Nguyen et al., 2014; Zygiel et al., 2017). Zygiel et al. (2017) have also described the likelihood that these mutations compensate for the replication defect afforded by the lacZα insert present in the M13 bacteriophage-based vector upon which the Ph.D.-7 (and Ph.D.-12) library was based. Thus, the particular peptide displayed (e.g., HAIYPRH, GKPMPPM, AKIDART) is merely a stowaway on a clone that propagates fast due to its gene II 5′-UTR mutation(s). In these clones, the peptide itself is completely arbitrary; it just happens to be the peptide displayed on a clone that picked up a mutation prior to or during library construction. As these mutations in the phage genome are unrelated to the displayed peptide, PhD7Faster 2.0 may not be able to predict this type of PrTUPs in the Ph.D.-7 library. In addition, Smith et al. indicated that the enhanced propagation rate of some PrTUPs may be due to the displayed peptide (Thomas, Golomb & Smith, 2010), and PhD7Faster 2.0 can be used to predict this type of PrTUPs in the Ph.D.-7 library. However, no direct evidence supports that displayed peptides allow the phage to propagate faster, and the biological mechanism remains to be further examined.

Conclusion

In this report, we propose an SVM-based tool, PhD7Faster 2.0, for predicting clones growing faster from the Ph.D.-7 phage display library. Ten-fold cross-validation results show that PhD7Faster 2.0 achieves an accuracy of 81.84% with 0.64 MCC and 0.90 AUC. The standalone version of the tool was also developed, which can predict PrTUPs within both traditional biopanning data and next generation phage display data. We also implemented a web-server for the proposed method, which can be freely accessible from http://i.uestc.edu.cn/sarotup3/cgi-bin/PhD7Faster.pl.

Supplemental Information

Supplemental Information 1 Supplementary Data.

Positive, negative and test datasets.

Click here for additional data file.

The authors are grateful to the anonymous reviewers for their valuable suggestions and comments, which will lead to the improvement of this paper.

Additional Information and Declarations

Competing Interests

Author Contributions

Data Availability

The authors declare there are no competing interests.

Bifang He conceived and designed the experiments, performed the experiments, analyzed the data, contributed reagents/materials/analysis tools, prepared figures and/or tables, authored or reviewed drafts of the paper, approved the final draft.

Heng Chen contributed reagents/materials/analysis tools, approved the final draft.

Jian Huang conceived and designed the experiments, authored or reviewed drafts of the paper, approved the final draft.

The following information was supplied regarding data availability:

Data generated or analyzed during this study are available in the Supplementary Data. The standalone GUI is at http://i.uestc.edu.cn/sarotup3/cgi-bin/PhD7Faster.pl and the source code can be downloaded from http://i.uestc.edu.cn/sarotup3/download.html.

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
