# Peer review of "PhD7Faster 2.0: predicting clones propagating faster from the Ph.D.-7 phage display library by coupling PseAAC and tripeptide composition"

_PeerJ, doi:10.7717/peerj.7131_

## Round 0.1 · original submission · Major Revisions

Both reviewers agreed that the manuscript is technically sound and describes an important tool. However, both of them noted a need to improve the manuscript. Could you please address the comments of the reviewers? Please flesh out your discussion of how/why propagation-related TUPs arise (gene II 5’-UTR mutations) and address the problematic linking of Pr-TUP enrichment to the displayed peptide sequence.

Reviewer 1 ·

Basic reporting

The manuscript “PhD7Faster 2.0: predicting clones propagating faster from the Ph.D.-7 phage display library by coupling PseAAC and tripeptide composition” by Bifang He, Heng Chen, Jian Huang proposed an SVM-based method to identify fast-growing peptides from the Ph.D.-7 phage display library, which is a update of the PhD7Fater tool. The SVM-based predictive model was further implemented into a web server, and a standalone version of the tool was also developed, which can predict propagation-related target-unreltaed peptides in small-scale biopanning data as well as large-scale phage display data. Minor revisions are required before publication.

Experimental design

The topic is important. Experimental design is correct.

Validity of the findings

The method is novel. Good results were obtained.

Additional comments

The manuscript “PhD7Faster 2.0: predicting clones propagating faster from the Ph.D.-7 phage display library by coupling PseAAC and tripeptide composition” by Bifang He, Heng Chen, Jian Huang proposed an SVM-based method to identify fast-growing peptides from the Ph.D.-7 phage display library, which is a update of the PhD7Fater tool. The SVM-based predictive model was further implemented into a web server, and a standalone version of the tool was also developed, which can predict propagation-related target-unreltaed peptides in small-scale biopanning data as well as large-scale phage display data. Minor revisions are required before publication.

I have a few small concerns:
1. The CD-HIT program is a widely used to remove redundant sequences. Did authors try to use it to remove redundant sequences in the benchmark dataset?

2. Page 6, "PseAAC and tripeptide composition have been widely used in protein prediction related research", authors should cite related references here.
Page 9, "Feature selection, thus, is a critical step to reduce feature dimensionality and build a highly effective prediction model", authors should also cite related references here.

3. I have noticed a few selected grammar/writing style errors (I assume someone else will proofread and edit the manuscript before publishing it).

·

Basic reporting

The manuscript, PhD7Faster 2.0: predicting clones propagating faster from the Ph.D.-7 phage display library by coupling PseAAC and tripeptide composition, by B. He, H. Chen, and J. Huang, reports a new web tool to predict propagation-related target-unrelated peptides (Pr-TUPs) from the Ph.D.-7 phage-displayed peptide library from New England Biolabs. The authors have made improvements compared to their previous version of the web tool (PhD7Faster 1.0) by employing the copy numbers of clones from the naïve library as well as the amplified library, by considering tripeptide composition, and by equipping the tool to handle large data sets such as next-generation sequencing data. PhD7Faster 2.0 is a valuable tool for researchers who perform phage display and/or are interested in target-unrelated peptides.

The manuscript is assembled in a professional manner, with sections appropriate for the content of the study. It is self-contained, and the amount of content is appropriate for one publication unit. It is organized, generally well-written, and conforms to professional standards of the scientific literature. Below I have made a few specific suggestions about wordings and grammar. I have also included here some small corrections to the citing of the literature (I have more significant comments about the relationship of this report to the literature in a section below).

Lines 39-40 Probably should have one or more references for the inorganic, organic, and biological materials at the end of the first sentence.

Lines 46-48 Does the Martins et al. reference apply to the whole sentence? If so, it should be at the end of the sentence. If not, there should be specific references for “materials science, biotechnology, pharmacology, cell biology and diagnostics.”

Line 49 The term “phage display screen” is a bit awkward. How about “The phage display methodology is notorious…”

Line 52 Not “which can fall” but rather “and can fall” because “which” is supposed to refer to the noun right before the comma (target of interest), but if you use “and” it’s clear that you are referring to the same subject of the sentence, the false positive TUPs.

Line 52 PrTUPs, as coined in Thomas et al., stands for “propagation-related” not “proliferation-related.”

Lines 52-53 Displayed peptides do not normally “react” with anything (this word implies a chemical reaction). They bind to targets and/or components of the screening system. (Reacting is a chemical process; binding is a physical process. Sorry I teach general chemistry! ;)

Line 56 lower growth rates

Lines 68-69 “archives special for” is unclear; I’m not sure what you’re trying to say here. Also, it’s not clear what you mean about “many different screening experiments” and “lack of target specificity” – are these two different ways that the TUPs are identified or are they the same/connected?

Lines 96-97 Better to write, “…to characterize both the naïve Ph.D.-7 phage display library and the same library after one round of amplification.”

Lines 173-174 “These measures were expressed as follows” implies that all five measures are going to be elaborated upon, but only the first four are actually defined. Either separate the AUC from the others in the list (line 172) or elaborate on it also.

Experimental design

Here I must insert the disclaimer that I am not a computational researcher, not do I know a lot about computational methods. I hope that the other reviewer was able to critically analyze the methods used in the study. For my part, I can say that the experimental design was clearly described and seemed to follow the conventions of the particular practices employed (at least based on the things I looked up while I was reading). It also appeared to me that the experiments were described in enough detail that they could be reproduced by another researcher who has a knowledge of computational methods. Additionally, the investigation was performed with high technical and ethical standards. In terms of the knowledge gap that is filled by this contribution, I would say that there is always more room for better methods to identify both selection-related and propagation-related TUPs. Phage display is a widely-used technique that can be complicated by the presence of target-unrelated peptides, which crop up in panning data and can be mistaken for actual target-binding peptides.

Validity of the findings

My primary concern about this article is the strong implication that propagation-related TUPs (Pr-TUPs) arise as a consequence of the displayed peptide, which I assume is the reason the authors employed peptide composition to develop the classifier (Lines 81-83, 110-144, 236-244). In fact, our three publications (Brammer et al., Nguyen et al., and Zygiel et al.) have demonstrated that numerous Pr-TUPS possess mutations in the 5’-untranslated region of gene II in M13 phage. Zygiel et al. describes the likelihood that these mutations compensate for the replication defect afforded by the lacZ insert present in the M13mp-based vector upon which the Ph.D.-7 (and Ph.D.-12) library was based. Thus, the particular peptide displayed (e.g., HAIYPRH, GKPMPPM, AKIDART, …) is merely a stowaway on a clone that propagates fast due to its gene II 5’-UTR mutation. In these clones, the peptide itself is completely arbitrary; it just happens to be the peptide displayed on a clone that picked up a mutation prior to or during library construction (see Zygiel et al., p.17, 3rd paragraph). When Matochko et al. showed sequence bias in both the naïve and amplified libraries, they did not suggest that the cause of clone enrichment was the displayed peptide. In fact, they clearly stated, “Our report uncovers ‘parasites,’ which do not have a specific amino acid sequence. Their high abundance cannot be predicted from positional abundance of amino acids … The biological mechanism that makes some sequences ‘parasitic’ is already known: they emerge due to mutations in the regulatory region of the phage genome” (and here they referenced Brammer et al). It was soon after this publication that we reported (Nguyen et al.) several more gene II 5’-UTR mutants that overlapped with the parasites Matochko et al. identified.

The authors support the use of tripeptide composition to develop predictors by citing how it was successful in predicting the subcellular localization of mycobacterial proteins (Zhu et al.) and apoptosis protein subcellular location (Liao et al.) (Lines 240-242). But these events are clearly dependent on the amino acid composition of the proteins involved, while the evidence around Pr-TUPS points to mutations that have nothing to do with the displayed peptide sequence.

Now, it is possible that PART OF the peptide sequence bias in the naïve Ph.D.-7 library is due to biological factors that make some displayed peptides more stable than others. It is likely that certain amino acids are censored because they create less stable peptides. But this censorship is a completely different phenomenon from that of displayed peptides making the phage faster, particularly after the library has been constructed and already contains the subset of peptides that are stable enough to have survived library construction. I would recommend that the authors focus on the biological basis for the amino acid censoring that may lead to the original sequence bias in phage display libraries, but avoid any implication that once the library is constructed, certain clones are enriched during amplification due to the properties of the displayed peptide (unless they can find actual experimental data that supports this). If the authors do intend to state that after the library is constructed, some displayed peptides confer to the phage properties that enhance infection, assembly, secretion (or whatever else), which consequently allow the phage to propagate faster, then they should (i) be clear about how in particular the displayed peptide affects these properties and (ii) back it up with some evidence from the literature.

Additional comments

Well-constructed paper overall, and very useful web-tool. Please flesh out your discussion of how/why propagation-related TUPs arise (gene II 5’-UTR mutations) and address the problematic linking of Pr-TUP enrichment to the displayed peptide sequence.

---

## Round 0.2 · Minor Revisions

Thank you very much for careful treatment of all suggestions made by reviewers. There are just a few minor points to address at this stage. First, may be beneficial to add material from your rebuttal letter to the article itself. Second, it may be worth to re-read the article (or give it to a colleague to read) to ensure the continuity of text flow.

Reviewer 1 ·

Basic reporting

Authors have made revisions according to my suggestion. I think the paper should be accepted.

Experimental design

The experimental design is correct.

Validity of the findings

The results are reliable.

Additional comments

The paper should be accepted.

·

Basic reporting

My general comments are the same as in my first review, and all the small edits that I recommended have been made. Good work!

One strange thing: This passage was mentioned in the author’s rebuttal, but I do not see it in the manuscript: “For example, the peptide HAIYPRH (a typical PrTUP) has been identified by 23 completely different targets according to results of searching the BDB database. The phage displayed the peptide was later verified to have a propagation advantage owing to mutations in the regulatory region of the phage genome (Brammer et al. 2008). HWGMWSY (a SrTUP) has been isolated by 10 completely different targets according to records in the BDB database. The peptide was proved to be a plastic binder (Vodnik et al. 2012), accounting for the target-unrelated selection of this peptide in multiple reported panning experiments.”

Experimental design

My comments are the same as in my first review.

Validity of the findings

The authors have added the key content that I requested, and I appreciate this improvement. However, I still have a few concerns about writing style and organization:

1. The new paragraphs are inserted at the very start of the discussion, making for a very abrupt start to the section. It also does not connect and integrate the new content to the pre-existing content. For example, the part about the use of tripeptide composition to develop predictors in the cases of the subcellular localization of mycobacterial proteins (Zhu et al.) and apoptosis protein subcellular location (Liao et al.) (Lines 262-265) is, in part, what warranted some discussion of whether peptide composition is the determinant of the property being analyzed (in those cases it is; in the case Pr-related TUPs, it is not). But the new content is not connected to the pre-existing content, which is 2-3 paragraphs later. Also, how does the censorship of peptides play into how this new method works? How does the presences of arbitrary peptides coincident with gene II 5’-UTR mutations play into how the new method works? I would recommend that the authors considered the organization of the first four paragraphs of the discussion with an eye for flow and cohesion.
2. The first sentence of each new paragraph is unclear. “Some peptides are likely to be censored...” The reader may think this means censored from the analysis, when it really means censored from being displayed on the phage in the first place. “The opposite, PrTUPs are biased towards during amplification.” This sentence is not grammatically correct. Also, “opposite” is a very specific word, which I wouldn’t use here. I think the authors may be struggling with these opening sentences, in part, because of my first point above about how the new content should be integrated with the pre-existing content.

Additional comments

N/A

---

## Round 0.3 · accepted · Accept

Thank you for making extensive changes to your paper. To improve the structure, I propose to make several additional changes (in the attached document) to further improve the flow and clarity.